# Impact of Social Capital on Health Behaviors of Middle-Aged and Older Adults in China—An Analysis Based on CHARLS2020 Data

**DOI:** 10.3390/healthcare12111154

**Published:** 2024-06-06

**Authors:** Zheyu Wang, Yong Fang, Xingwei Zhang

**Affiliations:** School of Public Health, Hangzhou Normal University, Hangzhou 311121, China; 2022111027011@stu.hznu.edu.cn (Z.W.); 2021111012063@stu.hznu.edu.cn (Y.F.)

**Keywords:** middle-aged and older adults, social capital, health behaviors

## Abstract

To actively respond to the challenges posed by population aging, people are paying more and more attention to healthy behavioral lifestyles, and the impact of social capital as an informal system on health behaviors cannot be ignored. This paper explores the impact of social capital on health behaviors of middle-aged and older adults based on 2020 CHARLS data. Using binary logistic regression models, we discussed the association between social capital and five health behaviors. The results suggest that structural social capital significantly increases physical activity and physical examination behaviors among middle-aged and older adults but also decreases the probability of abstinence behaviors. Cognitive social capital increases the probability that middle-aged and older adults will have a reasonable amount of sleep and physical activity. However, it also decreases the probability that smoking cessation behaviors will occur. Further attention needs to be paid to the role of social capital, the creation of a harmonious social environment and the enhancement of social trust, the strengthening of communities and grass-roots social organizations, and the provision of more platforms for the participation of middle-aged and older adults in social activities, to improve the quality of the healthy lives of middle-aged and older adults and, in turn, to promote the establishment of healthy behaviors.

## 1. Introduction

As China’s economy develops and industrialization accelerates, people’s lifestyles and disease spectrums change, and according to the National Assessment Report on Aging and Health in China published by the World Health Organization [1], the burden of disease is gradually shifting from maternal and child health problems and infectious diseases to chronic diseases. At the same time, with the acceleration of the “silver wave”, China has rapidly entered the stage of an aging society and has become the country with the largest elderly population in the world. According to the National Bureau of Statistics of China [2], by the end of 2022, China’s elderly population aged 60 years and above has reached 280 million, accounting for 19.8% of the total population, and the health problems of middle-aged and older adults are becoming increasingly prominent. China’s aging population is projected to increase the disease burden of chronic non-communicable diseases by at least 40% by 2030.

According to the health formula released by the World Health Organization, in maintaining human health, lifestyle and behavior habits account for 60%, environment accounts for 17%, genetic factors account for 15%, and health services account for 8%, indicating that a healthy lifestyle effectively affects people’s health. Among these behaviors, smoking, alcohol, exercise, and diet are significant determinants of health and are causally linked to cardiovascular disease, cancer, and many other chronic diseases [3]. Some data show that in developed countries in Europe and the United States, diseases caused by poor lifestyles lead to the death of 60–70% of the population, significantly affecting their quality of life and life safety. Therefore, it is essential to explore middle-aged and older adults’ health behaviors and influencing factors, as well as formulate targeted health policies and interventions to improve their health behaviors. 

Over the past decade, social and behavioral science theories have recognized the causal relationship between environmental factors and health behaviors, supported by empirical research results, and the correlation between social environment and health behaviors has been widely accepted as a significant determinant of health [4]. Sociologist Robert Putnam proposed the concept of social capital in the 1990s. Social capital is considered an essential factor for social development and individual well-being. Social capital refers to the connections between individuals or groups, namely social networks, reciprocity, norms, trust, and participation, from which people can obtain resources and help [5]. Social capital theory has had a broad impact in a variety of fields, with many studies confirming that social capital factors have a significant impact on health behaviors [6].

Developed countries in Europe and the United States began to study the impact of social capital on health behaviors in the 1990s, and it has been confirmed that social capital has a significant impact on physical activity, obesity, sleep [7], smoking [8], drinking, and other behaviors [9]. Evidence shows that trust, civic engagement, and social support are associated with drinking behavior [10,11], and Swedish and US adult studies have shown that physical activity and social engagement are positively correlated [12]. A Norwegian study verified that social capital reduces the likelihood of smoking, and both social participation and trust were negatively associated with smoking [13]. Health survey data in the United Kingdom found that alcohol consumption is related to sociodemographic variables. In contrast, social capital at the community level is not related to alcohol consumption, and only a high level of civic participation can increase residents’ alcohol consumption [10]. Yang L [14] found that social capital is related to sleep duration in the elderly, and a low level of social trust adversely affects sleep. Tarja et al. took health behaviors as an intermediary variable to study the impact of social capital factors (social support, social participation, and network, trust, and reciprocity) on health and found that social participation was correlated with all five kinds of health behaviors [15]. Using Bourdieu’s theory of capital, Xu et al. analyzed the relative importance of economic, cultural, and social capital and health behaviors in middle-aged and older adults [16]. However, due to the many perspectives on the measurement of social capital as well as the different methods of measurement, research results vary.

As far as the existing literature is concerned, researchers mainly analyze the correlation between social capital and health behaviors in the context of developed countries in Europe and the United States; at the same time, there are few studies on this subject in developing countries. Geographical and cultural differences may affect the relationship between social capital and health behavior. Especially in a large developing country such as China, the impact of social capital may vary across cultures, and more attention has been paid to the impact of social capital on individual health, with relatively limited research on social capital and health behaviors [17]. Therefore, exploring the impact of personal social capital and health behaviors has significant health implications.

This study is based on the China Health and Retirement Longitudinal Survey (CHARLS) data in 2020 to demonstrate the relationship between social capital and health behaviors. The main contributions of this paper are as follows. Social capital is closely related to the economic and cultural context, and the specific meaning and manifestation of social capital can vary between countries and regions. As a vast developing country, China has increasingly prominent problems of social and economic development and population aging. In the particular social background of China, it is of special significance to explore the impact of social capital of middle-aged and elderly groups on health behaviors. In addition, most studies have focused on the general population rather than middle-aged and older adults with higher health risks, which has implications for clarifying the impact of social capital on the health behaviors of middle-aged and older adults and the prevention of chronic diseases. In the special social context of China, focusing on middle-aged and older people with higher health risks, it is meaningful to construct structural social capital and cognitive social capital from the concept of social capital [18] and to explore the influence of social capital on the health behaviors of middle-aged and older people as well as the prevention of chronic diseases. 

## 2. Materials and Methods

### 2.1. Data Sources

This study used data from the China Health and Retirement Longitudinal Study (CHARLS). The tracking survey is a large-scale data collection project by the National Development Academy of Peking University and the China Social Science Survey Center of Peking University, specifically targeting middle-aged and older adults aged 45 and above. The purpose of the survey is to sort out micro data about the individuals and families of middle-aged and older adults in China. The baseline CHARLS survey was started in 2011, using the probability ratio scale sampling method (PPS), and is conducted every two years, covering 150 counties and 452 villages/communities in 28 provinces/autonomous regions/municipalities across the country [19]. This study selected data from the CHARLES database in 2020, cleaned the data as required, eliminated missing data on social capital and health behavior variables, and finally obtained 14,398 valid samples.

The CHARLS data are open and can be downloaded at the official website (https://charls.pku.edu.cn/ (accessed on 20 December 2023)) after registration and review.

### 2.2. Variable Measurement 

Health behaviors. According to Anderson [20], health behaviors refer to the adoption of relevant health practices by individuals that help maintain them. The health behaviors in this paper refer to the behaviors taken by individuals to prevent diseases and protect their health, including five specific dimensions, namely, “whether the sleep time is reasonable”, “whether not to smoke”, “whether not to drink alcohol”, “whether to exercise”, and “whether to have a physical examination in the past two years”.

In combination with the questionnaire, the first indicator is to select the average time of natural sleep every night (within the past month) asked in the questionnaire. The average sleep time of adults is more than 6 h per day, but in natural life, the sleep time of different individuals varies considerably. After excluding the outliers, the sample that slept between 6 and 10 h was defined as “1 healthy sleep”, and the sample that slept less than 6 h or more than 10 h was recorded as “0 unhealthy sleep”. For the second indicator, the question selected in the questionnaire was “Do you still smoke or have you quit smoking?”. The question options were ”1 still smokes”, “2 quit”, and ”3 never smoked”, with 1 recorded as ”0 smokes” and 2 and 3 combined and recorded as ”1 does not smoke”. For the third indicator, choose the question, “In the past year, have you drank alcohol, including beer, wine, or liquor?”, “How often do you drink?”. The question options are “1 drink more than once a month”, “2 drinks less than once a month”, and “3 drinks nothing”, the original 1 is redefined as “0 drinks more than once a month”, and the 3 is combined and re-coded as “1 drinks less than twice a month”. The fourth indicator selects three questions and options from the questionnaire: 1 “Do you usually do vigorous exercise (aerobic exercise, fast cycling, farming, heavy lifting) for at least 10 min continuously per week”, 2 “Do you usually do moderate-intensity exercise (cycling at a regular pace, mopping, tai chi, brisk walking, etc.) for at least 10 min per week”, 3 walks per week (for leisure, sport, exercise or recreation purposes). The three options are combined, and the residents who participate in any activity are recorded as 1. The residents who do not participate in any activity are recorded as 0, and the indicator of “daily exercise” is obtained. The fifth indicator, the question in the questionnaire, “When was the last time you had a routine medical examination since your last visit?”. After sorting out the samples that participated in physical examination in the past two years, they were recorded as 1, and the others were recorded as 0.

Social capital is “a social network, social trust, and social norms that can be enhanced through mutual cooperation” [21]. Some research suggests that social capital can influence health behaviors in various ways. The quantification of social capital has been a topic of academic discussion, and different scholars often have different insights and measurement methods, making it difficult to form a unified standard. This study divides social capital into structural social capital and cognitive social capital. Structural social capital focuses on the individual’s social relations and social activities, including social networks and social participation. In contrast, cognitive social capital focuses on the individual’s internal moral concepts and values and other spiritual elements. Based on previous studies and the availability of data, this study constructs social capital. Specific measures are as follows.

Structural social capital: This study takes participation in social activities that can improve social capital as an index to measure the level of social capital of middle-aged and older adults. By asking respondents, “Have you engaged in any of the following social activities in the past month?”, there are ten activities, including door visits, recreational activities, offering help, dancing exercises, club activities, volunteer activities, taking care of patients, attending training, stock trading, and surfing the Internet. If an activity is participated in, it is recorded as 1, and if it is not participated, it is recorded as 0. The cumulative score represents the degree of middle-aged and older adults’ participation in social activities. The greater the number of social activities, the richer the structural social capital. 

Cognitive social capital: The basic pattern of Chinese society is a family-centered differential pattern [22], particularly evident in rural society. With the rapid development of the market economy and the continuous advancement of urbanization, however, the mobility of middle-aged and elderly groups is lower than that of young people, and their social capital is still based on family ties. In this study, the perceived trust of middle-aged and older adults was used to measure their cognitive social capital. Respondents were asked, “If you need care in your daily life in the future, do you have relatives (other than your spouse) or friends who can help you in the long run?”. This question reflects the perceived trust of the respondents, with family and friends being the “first choice” of middle-aged and older adults to seek help when they are sick or in need of care [23]. Perceived trust in family and friends is a good measure of a person’s perceived social capital. In this study, the answer “yes” is defined as a high perceived trust and is recorded as 1; otherwise, it is 0. 

Control variable. This study controlled for age, sex (male = 1, Female = 0), educational level (illiteracy = 1, primary school and below = 2, secondary education = 3, higher education = 4), marital status (unmarried = 0, married = 1), place of residence (rural = 0, urban = 1), medical insurance (no = 0, yes = 1), per capita household expenditure, self-rated health status (very bad = 1, bad = 2, fair = 3, good = 4, very good = 5), and chronic disease (none = 0, some = 1). Table 1 shows the descriptive statistics for the variables. 

### 2.3. Statistical Analysis

Descriptive statistical analysis was used in this study, which mainly included the description and comparison of the basic characteristics and individual-level social capital variables as well as the dimensions of the study participants, as well as the current status of the health behaviors of middle-aged and older people.

Univariate analysis of variance was used to compare the health behaviors of middle-aged and older adults with different basic characteristics and different levels of social capital ownership. The chi-square test and t-test were used. The multifactor analysis is mainly used to analyze the influence of different basic characteristics and different degrees of social capital on the health behavior of middle-aged and older people.

Depending on the outcome variables, we used a binary logistic regression model to analyze the relationship between social capital and health behaviors. Specifically, the statistical model is as follows:$$logit(Pr)=\beta_{0}+\beta_{m}x_{m}+\beta_{n}x_{n}+\beta_{p}x_{p}$$

In the equation, Pr represents the probability of each health behavior. β_0_ is the intercept term of the model. x_m_ and x_n_ represent social capital’s social participation and trust variables. x_p_ is the set of control variables. Binary logistic regression was executed in this study using spss 26.0; *p* < 0.05 was considered statistically significant in this study.

## 3. Results

### 3.1. Descriptive Statistical Analysis

The demographic information of 14,398 samples is shown in Table 1.

### 3.2. Comparison of Health Behaviors of Older Adults with Different Characteristics

Basic characteristic variables and social capital variables that may affect the health behaviors of middle-aged and older adults are taken as independent variables, and the health behaviors of categorical variables are taken as dependent variables. A chi-square test or *t*-test is performed, a = 0.05.

The analysis results showed that different essential characteristics, age, sex, residence, marital status, education level, medical insurance, self-rated health status, income, chronic disease, social participation, and social trust had statistical significance on the health behavior of middle-aged and older adults (*p* < 0.05). The results are shown in Table 2.

### 3.3. Results of Binary Logistics Regression Analysis

Table 3 shows that both social capital are related to the health behaviors of middle-aged and older adults in China. When other variables were controlled, the structural social capital variable was measured by social activity participation, which was significantly correlated with drinking, exercise, and physical examination behavior. For each additional unit of social activity participation, the risk of alcohol consumption among middle-aged and older adults increases by a factor of 1.18 (OR = 0.849, *p* < 0.001), the probability of exercise will increase by 60.4%, and the probability of participating in a physical examination in the past two years will increase by 4.6%. Social trust was used to measure the cognitive social capital variable, and social trust was significantly correlated with sleep, smoking, and exercise behavior. For each additional unit of social trust, the probability of getting a reasonable amount of sleep increased by 18% (OR = 1.186, *p* < 0.001), the probability of not smoking increased by 10% (OR = 1.100, *p* < 0.05), and the probability of exercising increased by 60.4% (OR = 1.604, *p* < 0.001). 

According to the results, the influence of control variables is analyzed. The age factor is significantly related to the health behaviors of middle-aged and older adults. The older the age, the lower the probability of having a reasonable sleep time. The probability of smoking and drinking decreased with age, and the probability of physical activity and physical examination behavior increased. Gender factors have significant effects on the sleep time, smoking, and drinking behavior of middle-aged and older adults. Compared with men, women’s sleep time is more reasonable. The results of this analysis showed that men were much more likely to smoke and drink than women. People with higher education levels are less likely to smoke but more likely to drink. With the improvement in education level, the probability of exercise for middle-aged and older adults will increase. Marital status affects sleep duration and exercise behavior, and married people are more likely to have a reasonable amount of sleep than unmarried people. Middle-aged and older adults living in cities and towns tend to refrain from smoking, exercise, and physical examination. Middle-aged and older adults with medical insurance have a higher probability of physical exercise and physical examination. With the increase in personal income level, the probability of reasonable sleep time, active exercise, and physical examination of middle-aged and older adults increases. People who rated themselves in better health were more likely to get a reasonable amount of sleep but less likely to abstain from alcohol and have a physical exam. Chronic diseases have a significant correlation between sleep time, smoking, and exercise behavior in middle-aged and older adults. Middle-aged and older adults without chronic diseases have more reasonable sleep time, and middle-aged and older adults with chronic diseases have an increased probability of not smoking and active exercise. 

## 4. Discussion and Conclusions

Using data from the China Health and Retirement Longitudinal Study (CHARLS), this study focuses on the middle-aged and elderly population, dividing social capital into structural and cognitive social capital and exploring their effects on the health behaviors of middle-aged and older adults, respectively. After controlling for a series of variables, social capital is found to be a non-negligible and essential factor influencing the health behaviors of middle-aged and older adults.

This study shows that structured social capital, as measured by social participation, can promote active physical exercise and physical examination behaviors of middle-aged and older adults, consistent with previous research results [15,24]. High participation in social activities can provide middle-aged and older people with more social support and information resources. Through participation in social activities, older people establish and expand social networks, which can provide emotional support and motivate middle-aged and older people to participate more actively in behaviors such as physical exercise and regular medical check-ups [25]. Social participation increased the probability of drinking in this study, and one study found that social participation was associated with an increase in the number of drinking days per month [26]. The reason for this may be that drinking behavior is a social behavior, especially in those social environments where drinking culture is generally accepted [27], and middle-aged and older adults increase their drinking opportunities as they participate in social activities. 

The present study found that social participation in smoking was not significantly associated; it is possible that the effect of social capital on smoking behavior under different dimensions of measurement is not yet consistent; most of the studies on the relationship between social participation and smoking behavior in developed regions have found that smoking prevalence is higher in people with low social participation when compared to groups with high social participation [28]. Some scholars have concluded that social capital has a positive effect on smoking behavior and a facilitating effect [29]. This suggests that the impact of social participation on smoking behavior may be influenced by different cultural contexts, research methods, and measures and that further research needs to take these factors into account in order to more fully understand the mechanisms by which social participation influences smoking behavior. 

Second, we measured cognitive social capital using perceived social trust and found that social trust promotes rational sleep and exercise behaviors and reduces smoking in middle-aged and older adults. This finding is consistent with the results of most previous studies [29,30]. The reason may be that social trust affects the health behaviors of middle-aged and older adults through psychological effects. Higher social trust is conducive to relieving stress and spreading healthy behaviors, thereby reducing anxiety and pressure and improving sleep quality [31]. Older adults with higher trust are more likely to choose not to smoke [32]. In addition, family and friends affection can help middle-aged and older adults to carry out rehabilitation and health care activities and reduce loneliness [33], help middle-aged and older adults to control their smoking behavior, and supervise and promote the active participation of middle-aged and older adults in physical exercise. 

The above findings have significant research value. With the acceleration of the aging process in China, middle-aged and older adults often face the risk of chronic diseases, and by actively adopting health behaviors, the risk of illness can be reduced, the development of diseases can be delayed, and the quality of life of middle-aged and older adults can be improved [34]. In order to promote healthy aging, there is a need to place greater emphasis on the role of social capital and to improve the health behaviors of middle-aged and older adults. We make the following recommendations: (1) active aging has become a new concept to deal with aging, emphasizing that the elderly should face the old life with a positive life state. The community is the basic place of daily life for middle-aged and older adults in China and is one of the main places where people can participate in social activities. By mobilizing the strength of the community and grass-roots social organizations, we can provide more platforms for middle-aged and older adults to participate in social activities [35], improve the supportive environment for the social participation of middle-aged and older adults, and provide them with ways to access health-related information and resources. (2) Create a harmonious social environment and improve the level of trust in society. The government can use policy publicity and social media to advocate establishing a social environment of mutual assistance. (3) Promote the fine Chinese tradition of filial piety as the core and encourage children to give more spiritual and emotional comfort to the middle-aged and the elderly so as to improve the living conditions of the middle-aged and the elderly and to reduce the incidence of unhealthy behaviors. 

A review of the research on the relationship between social capital and health behaviors suggests that there are still some limitations in the existing research. The measurement of social capital has always been a controversial topic. This study uses data from the China Health and Retirement Longitudinal Study, and the questionnaire used is not specially designed for social capital questions, so the measurement of social capital has limitations due to the content of the questionnaire. To some extent, participation in social activities represents the social networks and social capital formed by middle-aged and older people in the community context and in the workplace. This study is based on 2020 data for analysis, and in the future, panel data can be applied to study the dynamics of health behaviors in the middle-aged and elderly population, and the use of panel data, combining both cross-sectional and temporal dimensions, can improve the precision of the estimates. This paper examines social capital at the individual level. It expands the measurement of social capital in the future to include the family, community, and even more macro levels of social capital.

## Figures and Tables

**Table 1 healthcare-12-01154-t001:** Descriptive statistics of the sample.

Variables	Definition	N (%)	Mean	SD
Age	Continuous variable	N = 14,398	61.77	9.041
Sex	Female = 0	7318 (50.83%)	-	-
	Male = 1	7080 (49.19%)		
Education	Illiteracy	2812 (19.53%)	-	-
	Primary and below	6738 (46.80%)		
	Secondary education	4553 (31.62%)		
	Higher education	295 (2.05%)		
Marital status	Not in marriage = 0	2002 (13.90%)	-	-
	Married = 1	12,396 (86.10%)		
Residence	Rural = 0	8538 (59.30%)	-	-
	Urban = 1	5860 (40.70%)		
Personal income	Continuous variable	N = 14,398	10.851	0.923
Health insurance	No = 0	755 (5.24%)	-	-
	Yes = 1	13,643 (94.76%)		
Self-rated health	Continuous variable	N = 14,398	3.17	0.983
Chronic disease	No = 0	3830 (26.66%)	-	-
	Yes = 1	10,560 (73.34%)		
Social participation	Continuous variable	N = 14,398	0.883	1.038
Social trust	No = 0	4684 (32.5%)	-	-
	Yes = 1	9714 (67.5%)		
Sleep time	No = 0	4860 (33.8%)	-	-
	Yes = 1	9538 (66.2%)		
Nonsmoking	No = 0	4735 (32.9%)	-	-
	Yes = 1	9663 (67.1%)		
Non-alcoholic	No = 0	4117 (28.6%)	-	-
	Yes = 1	10,281 (71.4%)		
Physical exercise	No = 0	1162 (8.1%)	-	-
	Yes = 1	13,236 (91.9%)		
Physical examination	No = 0	7613 (52.9%)	-	-
	Yes = 1	6785 (47.1%)		

Source: CHARLS2020.

**Table 2 healthcare-12-01154-t002:** Results of univariate analysis of health behaviors in middle-aged and older adults.

Variables	Sleep Properly		X2/T	Nonsmoking		X2/T	Non-Alcoholic		X2/T	Physical Exercise		X2/T	Physical Examination		X2/T
	No	Yes		No	Yes		No	Yes		No	Yes		No	Yes	
Sex															
Female	2777 (37.9%)	4541 (62.1%)	117 ***	1868 (25.5%)	5450 (74.5%)	365.29 ***	637 (8.7%)	6681 (91.3%)	2883.402 ***	589 (8.0%)	6729 (92.0%)	0.01	3843 (52.5%)	3475 (47.5%)	0.779
Male	2083 (29.4%)	4997 (70.6%)		2867 (40.5%)	4213 (59.5%)		3480 (49.2%)	3600 (50.8%)		573 (8.1%)	6507 (91.9%)		3770 (53.2%)	3310 (46.8%)	
Residence															
Rural	2903 (34.0%)	5635 (66.0%)	0.569	3037 (35.6%)	5501 (64.4%)	68.463 ***	2402 (28.1%)	6136 (71.9%)	2.185	826 (9.7%)	7712 (90.3%)	72.731 ***	5054 (59.2%)	3484 (40.8%)	336.146 ***
Urban	1957 (33.4%)	3903 (66.6%)		1698 (29.0%)	4162 (71.0%)		1715 (29.3%)	4145 (70.7%)		336 (5.7%)	5524 (94.3%)		2559 (43.7%)	3301 (56.3%)	
Education															
Illiteracy	1129 (40.1%)	1683 (59.9%)	73.871 ***	849 (30.2%)	1963 (69.8%)	18.92 ***	443 (15.8%)	2369 (84.2%)	329.45 ***	344 (12.2%)	2468 (87.8%)	119.224 ***	1566 (55.7%)	1246 (44.3%)	38.207 ***
Primary	2243 (33.3%)	4495 (66.7%)		2255 (33.5%)	4483 (66.5%)		2012 (29.9%)	4726 (70.1%)		561 (8.3%)	6177 (91.7%)		3629 (53.9%)	3109 (46.1%)	
Secondary	1407 (30.9%)	3146 (69.1%)		1553 (34.1%)	3000 (65.9%)		1527 (33.5%)	3026 (66.5%)		249 (5.5%)	4304 (94.5%)		2296 (50.4%)	2257 (49.6%)	
Higher	81 (27.5%)	214 (72.5%)		78 (26.4%)	217 (73.6%)		135 (45.8%)	160 (54.2%)		8 (2.7%)	287 (97.3%)		122 (41.4%)	173 (58.6%)	
Marital status															
Unmarried	872 (43.6%)	1130 (56.4%)	99.909 ***	613 (30.6%)	1389 (69.4%)	5.415 *	419 (20.9%)	1583 (79.1%)	66.914 ***	252 (12.6%)	1750 (87.4%)	63.944 ***	1033 (51.6%)	969 (48.4%)	1.522
Married	3988 (32.2%)	8408 (67.8%)		4122 (33.3%)	8274 (66.7%)		3698 (29.8%)	8698 (70.2%)		910 (7.3%)	11,486 (92.7%)		6580 (53.1%)	5816 (46.9%)	
Health insurance															
No	258 (34.2%)	497 (65.8%)	0.062	315 (41.7%)	440 (58.3%)	28.181 ***	221 (29.3%)	534 (70.7%)	0.179	85 (11.3%)	670 (88.7%)	10.913 *	455 (60.3%)	300 (39.7%)	17.461 ***
Yes	4602 (33.7%)	9041 (66.3%)		4420 (32.4%)	9223 (67.6%)		3896 (28.6%)	9747 (71.4%)		1077 (7.9%)	12,566 (92.1%)		7158 (52.5%)	6485 (47.5%)	
Chronic disease															
No	1006 (26.2%)	2832 (73.8%)	133.155 ***	1354 (35.3%)	2484 (64.7%)	13.569 ***	1220 (31.8%)	2618 (68.2%)	26.132 ***	325 (8.5%)	3513 (91.5%)	1.114	2121 (55.3%)	1717 (44.7%)	11.974 *
Yes	3854 (36.5%)	6706 (63.5%)		3381 (32.0%)	7179 (68.0%)		2897 (27.4%)	7663 (72.6%)		837 (7.9%)	9723 (92.1%)		5492 (52.0%)	5068 (48.0%)	
Social trust															
No	1772 (37.8%)	2912 (62.2%)	51.588 ***	1581 (33.8%)	3103 (66.2%)	2.363	1253 (26.8%)	3431 (73.2%)	11.557 *	522 (11.1%)	4162 (88.9%)	88.41	2522 (53.8%)	2162 (46.2%)	2.608
Yes	3088 (31.8%)	6626 (68.2%)		3154 (32.5%)	6560 (67.5%)		2864 (29.5%)	685 (70.5%)		640 (6.6%)	9074 (93.4%)		5091 (52.4%)	4623 (47.6%)	
Social participation	14,398		−3.275 *	14,398		−2.146 *	14,398		10.095 ***	14,398		−19.261 ***	14,398		−7.225 ***
Age	14,398		12.835 ***	14,398		−2.894 *	14,398		−5.094 ***	14,398		8.909 ***	143,98		−9.422 ***
Personal income	14,398		−8.667 ***	14,398		−0.451	14,398		6.321 ***	14,398		−11.329 ***	14,398		−6.161 ***
Self-rated health	14,398		−20.374 ***	14,398		2.404 *	14,398		11.969 ***	14,398		−0.004	14,398		1.899

Notes: *** *p* < 0.001, * *p* < 0.05; CI = confidence interval.

**Table 3 healthcare-12-01154-t003:** Correlation between social capital and health behaviors of middle-aged and older adults in China.

Variables	Model 1 Sleep Properly	Model 2 Nonsmoking	Model 3 Non-Alcoholic	Model 4 Physical Exercise	Model 5 Physical Examination
Social participation	0.997 (0.961,1.033)	1.034 (0.998,1.071)	0.849 *** (0.816,0.883)	1.604 *** (1.473,1.747)	1.101 *** (1.065,1.138)
Social trust	1.186 *** (1.100,1.279)	1.100 * (1.019,1.186)	1.001 (0.915,1.093)	1.604 *** (1.416,1.816)	1.046 (0.973,1.124)
Age	0.983 *** (0.979,0.987)	1.009 *** (1.004,1.013)	1.018 *** (1.013,1.023)	0.984 *** (0.976,0.991)	1.022 *** (1.017,1.026)
Sex	1.382 *** (1.281,1.491)	0.490 *** (0.454,0.528)	0.095 *** (0.086,0.106)	0.886 (0.776,1.011)	0.936 (0.872,1.005)
Primary and below	1.087 (0.985,1.199)	1.054 (0.951,1.168)	0.956 (0.837,1.092)	1.219 * (1.044,1.424)	1.044 (0.950,1.149)
Secondary education	1.137 * (1.018,1.270)	1.048 (0.936,1.173)	0.986 (0.856,1.135)	1.656 *** (1.369,2.003)	1.092 (0.983,1.213)
Higher education	1.213 (0.914,1.610)	1.414 * (0.531,0.942)	0.734 * (0.548,0.984)	2.202 * (1.061,4.572)	1.154 (0.892,1.493)
Marital status	1.217 *** (1.095,1.352)	1.054 (0.944,1.178)	1.030 (0.900,1.179)	1.333 ** (1.131,1.571)	1.061 (0.958,1.176)
Residence	0.942 (0.871,1.018)	1.304 *** (1.206,1.409)	0.951 (0.870,1.039)	1.367 *** (1.187,1.573)	1.739 *** (1.618,1.870)
Personal income	1.071 ** (1.026,1.118)	0.975 (0.933,1.018)	0.963 (1.020,1.013)	1.153 *** (1.075,1.236)	1.061 ** (1.019,1.105)
Health insurance	1.033 (0.880,1.212)	1.454 *** (1.248,1.693)	1.018 (0.848,1.223)	1.376 ** (1.082,1.750)	1.266 ** (1.087,1.474)
Self-rated health	1.373 *** (1.321,1.427)	0.988 (0.952,1.026)	0.858 *** (0.821,0.896)	0.961 (0.902,1.023)	0.965 * (0.931,1.000)
Chronic disease	0.800 *** (0.734,0.873)	1.115 * (1.026,1.212)	1.079 (0.981,1.187)	1.187 * (1.029,1.371)	1.054 (0.974,1.141)
Pseudo R2	0.069	0.047	0.298	0.080	0.046
N	14,398	14,398	14,398	14,398	14,398

Notes: *** *p* < 0.001, ** *p* < 0.01, * *p* < 0.05; OR = Odds Ratio; CI = confidence interval; and the 95% CI is shown in parentheses.

## Data Availability

The data of CHARLS2020 is publicly available at https://charls.pku.edu.cn/ (accessed on 20 December 2023) after registration and review.

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
