# Peer review of "Impact of Social Capital on Health Behaviors of Middle-Aged and Older Adults in China—An Analysis Based on CHARLS2020 Data"

_healthcare, 2024, doi:10.3390/healthcare12111154_

Round 1
Reviewer 1 Report
Comments and Suggestions for Authors
This study investigates the relationship between social capital and health behaviors of middle-aged and older adults through the data of the Chinese Longitudinal Survey of Health and Retirement of 2020. The manuscript is well-written and provides a clear context. The reference list cover the relevant literature adequately. The title is a somewhat closed statement about reflecting on the study and resembles a descriptive research title. However, the study is cross-sectional in nature and this analytical feature should be emphasized in the title (Relation, effect, impact e.g.). The research objectives and findings are summarized appropriately in the abstract section. Regarding the remaining sections, I have indicated below the topics that I believe will contribute to the development of the article.
The text between lines 108 and 110 is unnecessary and can be removed. The introduction section should be completed by writing a sentence indicating the purpose of the study. Between the 86th and 95th lines of the introduction section, the authors made some assumptions. However, the assumptions had not been supported by the reference.
In the Methods section, it is not explained how the parameter named "The self-rated health" is evaluated and what the min-max score value is.
Since the coding of the study parameters and the prevalence distribution is presented in Table 1, it would be appropriate to move this table to the results section.
Since the data presented under the heading "Basic information" in the Results section is a repetition of Table 1, it can be removed.
The title of Table 2 is insufficient to explain the table.
The authors said that for each additional unit of social activity participation, the probability of middle-aged and older adults not drinking alcohol will decrease by 15.1% (OR=0.849, P<0.001). Instead, it can be said that social activity participation increases the risk of alcohol use by 1.18 (=1/0.849) times. However, this statement describes the risk in a somewhat indirect way. Because logistic regression is used to calculation of a Odds Ratio which is related to the negative effects of independent variables on the dependent variable. When Odds Ratio smaller than 1, 1/Odds Ratio is used to calculate the risk.
Author Response
Dear Reviewer, Thank you for your consideration and constructive comments on our manuscript, which we have carefully revised, please see the attachment.

Reviewer 2 Report
Comments and Suggestions for Authors
This manuscript describes a study into the connection between social capital and health behaviours of middle-aged and older adults in China. While this is an interesting field of study, in its current form, there are three major issues with this manuscript that need to be resolved before it can be considered publishable.
1)
Methodologically, the use of binary logistic regression requires to abstract richer data from the CHARLS dataset such as the number of hours slept to binary variables (0 for less than 6 or more than 10 hours, 1 for 6-10 hours). While this might be a valid approach, the authors need to make a better job at arguing for this approach. Why not use multinomial/ordinal logistic regression instead?
2)
The manuscript claims that little is known about the connection between social capital and health behaviours for middle-aged and older adults in China. They seem to be unaware of recent work in the same context:
Xu, P.; Jiang, J. Individual Capital Structure and Health Behaviors among Chinese Middle-Aged and Older Adults: A Cross-Sectional Analysis Using Bourdieu’s Theory of Capitals. Int. J. Environ. Res. Public Health 2020, 17, 7369. https://doi.org/10.3390/ijerph17207369
In particular, Xu and Jiang's work connects indicators of social, cultural, and economic capitals with health behaviours such as smoking, binge drinking, stay up behaviour, exercising etc., i.e., the same or very similar concepts as in the current manuscript.
3)
The theoretical positioning should be clarified. Currently, the manuscript mixes a number of theories/schools such as Bronfenbrenner's socioecological theory (from the 1970s) and Putnam's social capital theory (from the 1990s). Closely related work such as Xu and Jiang's article mentioned above rely on Schneider-Kamp's health capital (from the 2020s), which incorporates social, cultural, and economic forms of capital. The authors would be well-advised to more clearly position their work in one of these three (or other relevant) theoretical frames. If you aim for combining multiple theories, please argue for this and explain to which degree these theories are compatible and what the opportunities and limitations of such a theoretical combination are.
Detailed comments:
P4 L127: "Anderson" needs a reference.
P4 L140-142: This seems like a duplication of the case of new respondents. Please fix.
Author Response
Dear reviewers, Thank you for your consideration and constructive comments on our manuscript, we have carefully revised the manuscript, please refer to the attachment.

Round 2
Reviewer 2 Report
Comments and Suggestions for Authors
The authors addressed the remaining issues satisfactorily.